# The unmet mental health needs of adolescents with HIV in eastern Tanzania: Experiences of healthcare providers, adolescents, and caregivers

Tasiana Njau[1,2,3]*, Bruno Sunguya[4], Dorkasi L. Mwakawanga[5], Agape Minja[1], Sylvia Kaaya[1], Abebaw Fekadu[2,3,6]

**1** Department of Psychiatry and Mental Health, Muhimbili University of Health and Allied Sciences, Dar es Salaam, Tanzania, **2** Centre for Innovative Drug Development and Therapeutic Trials for Africa (CDT-Africa), Addis Ababa University, Addis Ababa, Ethiopia, **3** Department of Psychiatry, School of Medicine, College of Health Sciences, Addis Ababa University, Addis Ababa, Ethiopia, **4** Department of Community Health, Muhimbili University of Health and Allied Sciences, Dar es Salaam, Tanzania, **5** Department of Community Health Nursing, Muhimbili University of Health and Allied Sciences, Dar es Salaam, Tanzania, **6** Department of Global Health & Infection, Brighton and Sussex Medical School, Brighton, United Kingdom

* tasiana2009@live.com

## Abstract

Adolescents with Human Immunodeficiency Virus (HIV) are at greater risk for mental health problems than their HIV-negative counterparts. However, there is a dearth of evidence on the need for mental health services, including interventions for depression in adolescents with HIV (AWHIV), in most low- and middle-income countries (LMICs). This study's objective was to explore the unmet mental health needs of AWHIV to inform the development and implementation of a psychological intervention for depression in AWHIV in Dar es Salaam, Tanzania. A descriptive phenomenological qualitative study design was used. Consultative meetings with providers and 45 in-depth interviews were conducted with AWHIV, caregivers, and healthcare providers (HCPs) to explore their experiences and unmet mental health needs for AWHIV. Data from the consultative meetings were triangulated to validate the obtained information with those from interviews. Data were organized and managed with the aid of NVIvo-11. The thematic analysis framework guided data analysis. Five major themes emerged: Experience of complex symptoms, unmet need for services, impact of the unmet needs, ways utilized in managing symptoms, and preferred intervention. Complex depressive symptoms expressed as physical, behavioral, or somatic complaints adversely affected ART adherence and academic performance, led to substance use, and compromised overall quality of life in AWHIV. HIV-Care and Treatment Centers (HIV-CTCs) did not conduct formal mental health screenings. Instead, caregivers and HCPs addressed the symptoms of mental health problems with death threats and corporal punishments. No evidence-based depression interventions existed in HIV-CTCs for observed symptoms. This study reports on unmet mental health needs with a clear impact on the lives of AWHIV, which may have significant implications for treatment adherence. There is an urgent need to

**Data Availability Statement:** All relevant data are within the paper and its Supporting Information files.

**Funding:** This study is part of a Ph.D. Project funded by the Centre for Innovative Drug Development and Therapeutics Trials for Africa (CDT-Africa), a World Bank Africa Centre of Excellence at Addis Ababa University granted to TN. The funders had no role in study design, data collection, and analysis, the decision to publish, or the preparation of the manuscript.

**Competing interests:** The authors have declared that no competing interests exist.

develop and implement effective and scalable interventions to address these mental health needs.

## Introduction

Adolescence involves many cognitive, social, psychological, reproductive, and behavioral changes and transition issues [1] Most mental health problems emerge during this period [2] affecting one out of every five adolescents worldwide [3, 4] Adolescents with HIV (AWHIV) experience greater mental health challenges than their peers in the general population due to their HIV status and related social and cultural factors [5–9]. These psychological and mental health challenges stem from difficulties accepting and coping with their HIV status, complicated grief following the death of a parent due to the disease, abuse, violence, and stigma and discrimination related to their HIV status [6, 7]. Consequently, these challenges increase their vulnerability to substance use, mood, anxiety, and cognitive disorders [10].

Depression is one of the most common mental health conditions faced by AWHIV [11, 12]. The burden of depression among AWHIV is higher among those in low-and middle-income countries compared to other settings [4, 13–15]. In Tanzania, depression affects 12–41% of AWHIV, three times the prevalence among adolescents in the general population [16, 17]. Depression affects ART adherence [16, 18] increases morbidity [19],mortality [20], caregiver burden, cost [18] and decrease quality of life [6, 21–26]. These negative consequences may relate to unmet needs that are not well understood and hence not addressed as per the global call for mental health services scale-up for adolescents with HIV [27].

Efforts to shed light on the mental health needs of these adolescents and ultimately develop interventions to address these problems are warranted [8]. Previous studies have emphasized the importance of understanding adolescents' insider perspectives when developing culturally responsive interventions [27–29]. However, only a few studies have qualitatively examined the experience of psychological and mental health problems of adolescents with HIV in Tanzania, [6], and none have examined their experience with unmet mental healthcare needs from the perspectives of the adolescents, their caregivers, and HCPs altogether. This study aimed to explore the subjective experiences and unmet mental health needs of AWHIV to inform the development and implementation of psychological intervention for depression in AWHIV in Dar es Salaam, Tanzania [30]. In addition, we included the perspectives of caregivers and HCPs, as their understanding of the mental health needs of adolescents with HIV may affect both the process and outcomes of service delivery.

## Material and methods

### Study design

We employed a descriptive phenomenological qualitative study to explore the experiences of adolescents, HCPs, and caregivers and identify unmet mental healthcare needs among adolescents with HIV. A qualitative research design was appropriate for this study as the experience is a real phenomenon involving social processes.

### Study context

The study was conducted in Dar es Salaam city, located in the eastern zone of Tanzania's mainland. The region is divided into three districts: Temeke, Kinondoni, and Ilala. The study took

place in Kinondoni district, a densely populated district with a higher (5.6%) HIV prevalence than the national prevalence (4.8%) [31]. Seven Primary HIV care and treatment centers provide services to many adolescents in the district. Data were collected in three of the 7 highly dense clinics purposively selected with the help of The Management and Development Health (MDH), based on active engagement in delivering adolescent care. All three sites offer adolescent-friendly services, which means they have separate clinic days and packages of services for adolescents. MDH is a not-for-profit organization that collaborates with the Government of Tanzania to implement HIV healthcare and treatment services for the community. It supports the provision of adolescent-friendly services in these centers.

## Study participants and recruitment

The study engaged 45 participants composed of 15 HCPs, 15 adolescents, and 15 caregivers. Adolescents and young persons aged 11 to 24 who have disclosed their HIV status were included. For HCPs, nurses and physicians with at least two years of experience providing care and treatment to adolescents with HIV were included. The study also included caregivers (parents or guardians) of adolescents receiving HIV care and treatment in the three selected clinics but not necessarily that of adolescents who participated in the study. Only those who did not consent or were mentally or physically unable to participate were excluded from the study.

To gain greater insights into participants' experience of mental health needs and services, we utilized a maximum variation purposive sampling approach [32] to recruit a diverse sample of participants from different clinic sites. Adolescents who can communicate well were selected with the assistance of a healthcare provider familiar with the adolescents. An equal number of older and younger adolescents were included. Similarly, each selected clinic's personnel in charge assisted in recruiting targeted HCPs. HCPs were selected based on having more experience or expertise in the dynamics of mental health problems in AWHIV. Those with more knowledge of the Tanzania health system and mental health services provision were given a higher opportunity to participate in the study.

## Data collection guides and procedures

Data for this study were collected between May and June of 2021. For the in-depth interviews, different semi-structured interview guides (for HCPs, caregivers, and adolescents) were developed in English and then translated into Kiswahili. The guides were developed based on the existing literature [6, 8, 16, 33] and the researchers' expertise in mental health. The guides consisted of open-ended questions and probes that explored information regarding experience with adolescents' mental health needs, experience with caring for adolescents, and their needs for HCPs and caregivers. To ensure data quality, three research assistants with a bachelor's degree, fluent in English and Kiswahili, who had received mental health training and experience in collecting qualitative data, were recruited. The data collectors received training on the study's objectives, the guides, informed consent, and the research process.

All interviews with adolescents and caregivers took place in respective CTCs, in a private and quiet room chosen by the healthcare provider. HCPs were interviewed in their offices at each clinic. Two consultative meetings were conducted with three HCPs and two facility in-charges to confirm the accuracy of the information obtained during in-depth interviews. One of the consultative meetings was done via ZOOM meetings due to COVID-19 situations in the country at the time of data collection.

Written informed consent was obtained from participants after explaining the purpose of the study and that the session would be audio-recorded. All interviews were conducted in Kiswahili, audio-recorded, and lasted between 30 and 45 minutes. For each discussion, additional

notes were taken to supplement the audio-recorded information. Interviews were conducted until saturation occurred for each group of participants (adolescents (12 interviews), caregivers 9 interviews, and HCPs 13 interviews); when no new information was obtained from participants and redundancy was reached [34, 35]. However, Since the actual number of interviews at which saturation occurred was close to what was approved by the Ethics board (45 interviews), we continued to 15 per category to check if new information will arise.

## Data analysis

Audio recordings were transcribed verbatim in Swahili and then translated into English. Before coding, the research team cross-checked the translation accuracy and completeness of the transcripts against the original scripts. Three authors (TN, DM, and AM) utilized a deductive-inductive team-based approach for codebook development. Thematic analysis following procedures described by Braun and Clarke guided the analysis of the data [36]. The analysis was aided by the qualitative software NVivo-11. Each transcript was coded independently until no new code/ insights emerged. Next, the three authors discussed the codes and emerging themes from the transcripts. During the coding process, unresolved discrepancies and disagreements with the codes or coding process were discussed and resolved with the team. A final codebook was compiled from the agreed codes and code definitions. Sub-themes and themes were then generated from the general list of codes based on their relationships. Themes were reviewed and discussed by the researchers (TN, AM, DM, SK, and AF), discrepancies were noted and identified themes were finalized and presented in the result section.

## Ethical considerations

Ethical approval to conduct this study was obtained from the Addis Ababa University Institutional Review Board (IRB), Ethiopia (Ref. No. 051/20/CDT) and from the Muhimbili University of Health and Allied Sciences IRB in Dar-es-Salaam Tanzania (Ref. No.DA.282/298/01.C/ 053). The Dar es Salaam Reginal Medical officer, Kinondoni District Medical Officer, and incharges from each clinic granted permission to collect data. Written informed consent for participation and a record of the interview were sought from all study participants aged 18 and above. A parent or guardian provided written permissions for adolescents under 18 years of age, with the adolescents giving assent. Those who participated via phone call or ZOOM meetings had consented in a previously physically held interview. They were informed of the possibility of being contacted if clarification or additional information was required. The study also adhered to the ethical implementation of research activities during the COVID-19 pandemic guidelines by following the government's directives on preventing the transmission of COVID-19 to study participants, healthcare workers, and fellow staff.

## Results

The overall sample consisted of an equal number of adolescents, caregivers, and HCPs (n = 15 for adolescents, n = 15 for caregivers, and n = 15 for providers).

### Adolescents

There were eight female and seven male adolescent participants. Their age ranged from 12 to 19 years, with a mean of 15.2 years. Their period in HIV care and treatment ranged from three years to twelve years, with a mean of 6.8 years. Eight were in primary school, and 7 in secondary school.

## Healthcare providers

Eight of the providers were clinicians with a tertiary level of education, and seven were nurse counselors. Five had a diploma in nursing, and two had secondary education with additional training in community health and HIV counseling. The majority (n = 13) of the HCPs were females. Clinicians' ages ranged from 25 to 53 years, with a mean of 33 years. The ages of Nurse Counselors ranged from 27–64 years, with a mean of 47.8 years. Experience working in HIV care and treatment centers ranged from 3 to 15 years for nurses and 2 to 7 years for clinicians.

## Caregivers

We included twelve women and three men who had a mean age of 45.6 years (range 27–61) and lived with adolescents with HIV as a parent (n = 6) or primary caregiver (n = 9) for a mean period of 9.3 years (range 3–15 years). In addition, nine possessed a primary level of education, five had a secondary level of education, and one had never been to school.

The analysis of the responses revealed five main themes related to the experience and mental health needs of AWHIV in Kinondoni. These include Experience of complex symptoms, unmet need for services, impact of the unmet needs, ways utilized to manage symptoms, and preferred intervention. Table 1 below summarizes these key findings.

## Experience of complex symptoms

Participants reported several psychological/mental health symptoms as cognitive, emotional, behavioral, or somatic. Adolescents reported not fully understanding how best to express these symptoms to others. Still, they believed the problems made them easily irritated, look different from others, constantly tired, unable to perform tasks the way they wished, and have suicidal thoughts, as described by this 17years old female adolescent.

**Table 1. The experience and unmet mental health needs of AWHIV in Dar es Salaam.**

| Themes | Sub-themes |
|---|---|
| Experience of symptoms | Cognitive symptoms |
| | Emotional symptoms |
| | Behavioral symptoms |
| | Physical/ Somatic symptoms |
| Unmet needs for services | Inadequate management of psychological/mental health problems |
| | Insufficient support from providers |
| Impact of the unmet needs | Poor medication adherence |
| | Poor academic performance |
| | HIV and substance use risk behavior |
| | Physical health related effects |
| Ways utilized to manage symptoms | Scolding and punishment |
| | Use of traditional remedies |
| | Spiritual beliefs and prayers |
| | Change of environment |
| Preferred Intervention | Education related to psychological/mental health |
| | Sustainable psychological support |
| | Integration of mental health services within HIV care and treatment facilities |

*". . .. Most of the time, you remain silent. You fight the situation alone. You can't explain because you do not know how. The only thing they see is that you look different, or maybe you are easily irritable and not doing your activities as required, but most of the time, you are fighting the constant need to die and not being able to stop them[thoughts]".*

The description of experiencing mental health problems included a sense of constantly carrying a heavy load, thinking too much, having many thoughts, and living alone in a "dark room," as described by this 19-year-old boy.

*"It feels like carrying a heavy load on your head . . ..and pressing your shoulder constantly. . .it is very hard. Like living in a dark room, you have so many thoughts, and you can't stop thinking because you are different from them; no one will understand. You just stay alone".*

Caregivers perceived psychological problems, specifically "overthinking" and anger in adolescents due to being different from others due to their HIV status, mainly because other family members were HIV-negative. They also described symptoms that they associated with psychological/mental health problems in the adolescents they live with, including looking sad/immersed in thoughts, difficulties with anger, sleep, appetite, isolation, and low energy and motivation, which they referred to as laziness, as described by a parent with primary education.

*". . .He tends to be very lazy. . .angry and so harsh even for a minor thing or situation. . .he just gets angry. . .food; he will not eat; he will leave it there. He has so much need to sleep. He sleeps a lot."*

Similarly, HCPs reported adolescents having many thoughts, looking sad, poorly communicating, not mixing with others, having suicidal ideation, and crying. They believed the positive HIV status subjects adolescents to psychosocial challenges that lead to these symptoms. With 4-year experience working with adolescents with HIV, this clinician described some symptoms, including physical complaints. She said,

*". . .Some of them may be mute and not talk to you, some cry a lot. When you are lucky that they talk, they say they have no reasons to live, or life has no meaning. For the younger ones, if they feel like you are disturbing them with questions, they tell you they have a headache, chest pain, or any other body part, but everything comes out normal when you order an investigation. In short, these kids reach a point when they lose hope to continue living. I can say they are very depressed."*

## Unmet needs for services

None of the clinics reported conducting a formal screening for mental health problems. However, they reported offering counseling and advice to address psychological or mental health symptoms that adolescents present with. However, there were no formal sessions, and the content was said to be what the provider felt appropriate to the presented problem of the adolescents. They reported informally asking about mental well-being, for example, probing about what might be bothering them if they see them crying, not taking their medication, or when parents complain of behavior change.

*"There are no guidelines on identifying or managing psychological or mental health problems. . .it's up to you as a provider to think outside the box when a parent complains. We mostly do adherence counseling and tell them not to overthink if you notice them crying, . . .but honestly, there is not much we can do" (Clinician, 5-year experience)*

On the other hand, adolescents also reported a lack of mental healthcare and not feeling understood by HCPs. While some participants reported having never asked for help, because they did not know where or how to ask for it, some reported help-seeking. Those who did ask for help said not to have felt understood by the healthcare provider and that the counseling they receive were not helpful, as narrated by this 19-year-old adolescent.

*"I explained my problem to him (having many thoughts, feeling different, anger, and a constant death image), and he told me, okay, take your medication on time, and avoid overthinking. . .aah, how do you prevent them [thoughts]? It's difficult. . .you can't; it didn't help".*

Adolescents consistently reported that the advice or the counseling offered had nothing helpful in dealing with psychological or mental health problems. Adolescents commented on how caregivers and HCPs use threats of their death to intervene in what they may see as a problem. This 18-year-old boy narrated his experience of having nowhere to seek help for psychological problems that interfered with him taking his ART medication and how it impacted care-seeking.

*"At home, you get shouted at and threatened. . .you come here [at CTC], it's just being scolded and threatened about death. Today you talk to the doctor, tomorrow with a nurse, the other day to someone else, but no one understands you; you better keep quiet and not share your problems".*

## Impact of the unmet needs

Psychological/mental health symptoms were reported to impact engagement in HIV care, academic performance, social interaction, and overall quality of life. Adolescents said that psychological problems were one reason they would not take their medication. Persistent symptoms (feeling sad, alone, worthlessness, loss of interest, and being different (due to HIV) and the associated loss of will to live meant that they see no reason to continue taking the ART medication. They described having intentionally stopped using their antiretroviral treatment as a means of suicide.

*". . .I thought the easier way to die was to stop taking medicine. . .. it was better to die. I overthink and don't sleep at night; just think and cry myself to sleep . . .you know if you do not take medicine, you will die soon". (Adolescent, 19 years old).*

Poor academic performance and impaired social interaction were noted by participants to be among the impacts of poor mental health. Adolescents reported experiencing difficulties with memory and concentration, which interferes with their academic performance and exacerbates other feelings like sense of worthlessness and suicidal thoughts, as this 18-year-old narrates.

*". . . I feel alone . . .even when I learn, I don't understand. I fail! After all, why should I study. . .I am not worth anything. . .I get upset so much.. . .no one understands that it is so*

*hard to concentrate on the book when you constantly think of how worthless you are. . . .not only do you despair, but you also want to die. That's why you don't even take medicine. What are they for when the only thing they do is keep you suffering a miserable life?"*

On the other hand, caregivers also reported that symptoms like sadness, isolation, losing hope, anger, and loss of interest in seeing their friends negatively affect academic performance and make it difficult for adolescents to take ART medication.

*"Taking her medicine becomes a problem; she looks like someone in deep thought and becomes weak. . . .she misses school, and her performance is poor because she is not interested in study-ing. . .. She no longer visits her friend (Caregiver, secondary education)."*

Furthermore, mental health problems and the associated symptoms like anger and despair were perceived by participants to increase sexual risky behavior and lead to the intentional transmission of HIV as this caregiver with a secondary level of education and lived with AWHIV for five years explains.

*"It is the despair, and they seem to be constantly angry. She knows her HIV status, and she gets a partner and does not use protection . . . it is like, let me do it, so he also gets HIV. If it is true that I have HIV, many people must be infected, too; she is determined to infect others; I think it is due to the emotional problems that she feels it's okay to infect others too."*

Providers from all clinics perceived suicidality and non-adherence to be one of the signifi-cant impacts of mental health problems in AWHIV. Non-adherence was defined as not using ART medication as prescribed and failure to comply with clinic schedules as planned. This was reported as a challenge, mainly due to medication resistance that led to the need for sec-ond-line ART treatment. This nurse counselor with 15 years of working experience in HIV care narrates,

*". . .. Adolescents stop taking medication, and for that reason, most of them are on second-line treatment."*

Like caregivers, HCPs perceived psychological and mental health problems to predispose adolescents to the use of substances and engagement in sexual risk behaviors. Caregivers were more likely to report adolescents engaging in groups that predispose them to substance-related risk behaviors [the use of alcohol, cigarette, and cannabis], while HCPs perceived both sub-stance and HIV and sexual risk behavior.

*". . . Some decide to drink alcohol, smoke cannabis, join bad groups, and engage in prostitu-tion." (Nurse Counselor, 7-year experience)*

## Ways utilized to manage symptoms

Caregivers thought the symptoms were stereotyped behavioral problems and failure to accept their HIV status. The despair of HCP and caregivers struggling with managing mental health-related symptoms in adolescents is evidenced by using punishment to help the adolescents. Caregivers, for example, reported that HCPs help them discipline their children when they show behavioral symptoms (e.g., lack of motivation to study) or do not adhere to medication by warning them about the effects. Punishment (caning) and scolding ("Kumsema") were one

of the ways used by caregivers to manage symptoms like poor appetite and crying for no apparent reason, as stated by this caregiver with secondary education who described how she dealt with behavioral symptoms in a 14-year-old adolescent LWHIV.

> *"Oh (yes), the nurses used to warn her whenever we came here. They say. . . why don't you want to eat or take your pills? I also bought appetite-boosting medication, but it has not helped; the stick has helped; she is now eating when threatened or spanked with a stick. Her father helps with this. . .. spanking with a stick (caning) also helps when she misbehaves or shows anger tantrums for no apparent reason".*

Contrary to that, Adolescents perceive caregivers' punishment and providers' harsh verbal warnings as one of the maintaining factors for their problems. This 15-year-old boy describes his experience with corporal punishment.

> *"They make the problem bigger when they do not understand and beat you. You get tired, and no one understands. . . and if you don't work, you get hit. . . there was a day my uncle beat me so much because I could not eat. . . I was in so many thoughts and not feeling hungry. . . . that was the night I poisoned myself, but sadly, I ended up at [hospital]".*

Caregivers reported using remedies, utilizing social support, and offering gifts to adolescents. They also use prayers even though they were unsure if church rituals, like "stepping on Holly oil," could remove adolescents' psychological /mental health symptoms.

> *". . . It is just making efforts to attend church and pray. She goes on the Holy Communion day and treads on anointing oil even though we are unsure if people are getting healed as we are made to believe". (Caregiver, Primary education).*

Caregivers further perceived the need to be educated about ways to help adolescents with psychological/mental health problems. The need for mental HCPs was also cited as this caregiver with primary education described her experience living with adolescents with mental health problems and HIV.

> *"It is very stressful for a parent who is raising these adolescents. . .. We need to know how to help. . .when she is sad, and I also have many things that confuse me. I find myself getting more confused . . . it is not enough. We need an expert for our children".*

## Preferred intervention

Participants recommended psychological services, and preferred them over medication for mental health problems. Consistently participants suggested psychological services delivered within the HIV care and treatment centers and offered in individual sessions as the issues were perceived as "too private to be discussed in a group." Adolescent participants recommended that existing youth clubs be improved and expanded to offer mental health knowledge and information. This 16-year-old adolescent describes his preference for psychological and mental health problems.

> *"We should not be taking medication every day [during appointments]. If possible, we should also be given appointments for psychological services. . .you know these problems are too private. You cannot share them in those clubs or groups. The clubs may be general things like*

*dealing with psychological problems; they would be more helpful than repeating the same things we already know".*

Similarly, caregivers thought psychological intervention to address common psychological and mental health problems would be an excellent opportunity for adolescents. It was consistently reported that HCPs should include psychological intervention within the care package within CTC, and education and awareness programs should be provided in peer-led clubs. Participants also recommended a clear schedule so adolescents may understand when, where, and who provides that service and how to request or access it within HIV care and treatment facility.

*There should be a special HCP who provides psychological counseling and should set dates like how they are given appointments for taking pills. It will be easy if our kids know who and which room to go for that service when they come" (caregiver, secondary education)*

On the other hand, HCPs recommended training on assessing and managing common mental health problems in AWHIV. Their experience indicates that not knowing how to help adolescents with depressive symptoms has left them helpless and feeling of having nothing more to give. They believed the training would increase their confidence and give them skills and competency to address common mental health problems and associated challenges. With six years of working experience as a nurse counselor for adolescents with HIV, this nurse had this to share.

*"The problem here is not seeing the symptoms . . . we see them, we try to help them where we can, but the truth is, we often don't know what to tell the young person. . . we do not have the expertise. You feel helpless and tired since you have nothing to offer. You find a young person sick every day because he doesn't even take his medication as directed. . .. he has psychological issues, and some contemplate suicide; if you look at yourself [as a provider], you don't know where to start. . .. We wish we could get training on what can be done apart from our routine. . .we need to do that to save our children".*

## Discussion

This qualitative study explored in-depth how adolescents with HIV, HCPs, and primary caregivers experienced unmet mental health needs and its implications in adolescents with HIV. To our knowledge, this is the first study in sub-Saharan Africa that examined the multiple experiences of adolescents with HIV, their caregivers, and HCPs. While the broader understanding of psychological and mental health problems identified in this study appears consistent with other studies of adolescents in Tanzania [37], the findings from our research shed light on the unmet mental health services needs and add a new dimension to the larger picture from multiple vantage points.

### Experienced symptoms

The symptoms experienced were complex, with a combination of cognitive, behavioral, and somatic symptoms that were difficult to explain. However, the pervasive negative and emotional nature of the symptoms are suggestive of depression [38]. In this study, adolescents with HIV experience depressive symptoms predominantly expressed physically and manifested as behavioral problems or somatic complaints. In Tanzania, these are considered culturally

acceptable expressions of emotions and psychological distress. Participants' descriptions of having too many thoughts are similar to other studies of depression in SSA [39].

Adolescents, their caregivers, and HCPs implicated the HIV in the experience and expression of depressive symptoms, particularly the difficulty of accepting having the condition as a significant challenge affecting the mental health of these adolescents. Similarly, Dorothy and colleagues [6] have reported that more than half of AWHIV had difficulties of accepting their HIV diagnosis.

## Unmet services need and ways utilized to manage symptoms

The study finds a remarkable level of unmet need for mental health services among AWHIV. Participants reported a complete absence of mental health services within HIV care and treatment facilities in Kinondoni Dar es salaam, Tanzania. There is a lack of understanding of both the contextually relevant factors that influence how adolescents experience depression or respond to help, and how to offer help to adolescents both among caregivers and HCPs. Thus, caregivers and care providers resort to unhelpful and sometimes dangerous interventions to deal with the depression-related symptoms and behaviors of the adolescents. For example, while caregivers use harsh verbal and physical punishments, care providers appear to collude or at least ignore the concerns of adolescents. Similar to the findings of a trial by Betancourt and colleagues [40], adolescents in this study reported that harsh physical or verbal discipline increased depressive symptoms. HCPs go against the expected norms of empathy and adherence to the highest professional ethics and standards and resort to harsh verbal gestures to address these symptoms/behavioral problems. The decline in empathy interferes with communication, decreases the adolescent's satisfaction, and interferes with the helping relationship [41] and quality of care in general. Although illustrates the complexity of delivering mental health services to adolescents with HIV, it is also a reflection of the failure of the health system. It is unfair on HCPs to expect them to deal with problems when their training has not equipped them with the knowledge and skills required to manage Depression.

## Impact of the unmet needs

Participants perceive those depressive symptoms hinder adolescents' engagement in HIV care, academic performance, social interaction, and overall quality of life. These may be attributed to the nature of depression and its functional impairment. In a study conducted in Kilimanjaro, Tanzania [16], almost a quarter of adolescents did not adhere to ART Medication, which was associated with mental health symptoms. The current study suggests that depression should be considered one of the factors associated with non-adherence to antiretroviral therapy among adolescents with HIV, as was reported elsewhere [7]. One of the alarming findings was that, for these adolescents with HIV, not taking their ART medication was a means to die. Other studies have also reported that adolescents defaulting to the ART treatment aimed to achieve "a slow death" [40].

Moreover, depressive symptoms were also believed to increase sexual risk behaviors and substance use. Confirming what has been extensively documented in the literature among adolescents [42, 43] partly as a way of coping with depressive symptoms [43].

It was found that adolescents, HCPs, and caregivers are all affected by the effects of depression. While adolescents are overwhelmed by depressive symptoms and do not feel understood by caregivers and HCPs, caregivers are also overwhelmed by the signs and behavioral manifestations of depression. Caregivers struggle unsuccessfully to find assistance in a biopsychosocial and spiritual context. For HCPs, the lack of mental health knowledge and skills needed to

address depression in adolescents caused burnout, defined as decreased sense of personal accomplishment and emotional extortion.

## Preferred intervention

Participants in this study preferred psychological interventions offered by HCPs in individual sessions within HIV care and treatment facilities. Brief psychological interventions, in addition to addressing depression, may solve the practical problem of low adherence rates to medication in this age group. Brief psychological interventions are recommended in the mental health gap action program intervention guide and other guidelines as first-line treatment for common mental disorders like depression [23], especially in adolescents.

It is not surprising that adolescents prefer individual sessions over group sessions for psychological intervention. Diverse Eastern and Southern African studies indicate initial evidence that group-based psychological interventions for adolescents lack acceptability feasibility, resulting in poor efficacy [44–46].

The preference is also for care to be provided within the HIV care system. In line with this preference, and for other pragmatic reasons, for example, the low number of mental health specialists, integrating mental health services in all HIV care and treatment centers that provide adolescent care is important. This is central to providing comprehensive services to adolescents with HIV [17, 47–49]. Along with this, there is also a need to develop an integrated system of care that links adolescents to specialized mental healthcare they need when they need it [10, 24, 27].

## Strength and limitation

The report of this paper followed the Consolidated criteria for reporting qualitative studies (COREQ): 32-item checklist [50]. The study included caregivers who play an essential role in the early recognition of adolescent mental health symptoms but are typically overlooked. The study achieved sufficient information power due to the inclusion of three groups of participants. However, it was difficult to discern whether the information provided by participants was accurate. We worked to minimize this by comparing opinions with those of other respondents and holding a meeting of HCPs to check the veracity of some of the opinions. Using bracketing (probing the same questions differently and being naive on the topic) during data collection, multiple coders to interpret data, and triangulated data sources minimized researchers' bias. Some participants were guarded during interviews, especially when discussing negative experiences with service delivery, out of fear of retaliation from the services. We resolved this by ensuring confidentiality throughout the interview and using interviewers not involved in delivering their HIV care services. Quantitative data could have strengthened the findings of this study. However, the information obtained in this study was sufficient to inform the development of a brief psychological intervention for depression, which was the ultimate aim of the study.

## Conclusion

A relatively great burden of distressing symptoms and unmet mental health needs among AWHIV increases the burden on caregivers, induces burnout among HCPs, and results in poor treatment outcomes. Depressive symptoms cannot be ignored as they are associated with high levels of distress, functional impairment, risky behavior, and suicidality. Depression and other common mental health problems must be fully addressed and integrated into HIV care for adolescents. Developing and implementing a brief psychological intervention that will be

integrated into HIV-CTC in Tanzania to manage depressive symptoms and enhance ART Medication adherence among adolescents with HIV should be prioritized.

## Supporting information

**S1 File. In-depth interview guides.**
(DOCX)

**S2 File. Consolidated criteria for reporting qualitative studies (COREQ): 32-item.**
(DOCX)

## Acknowledgments

We appreciate the support from the administrative offices in the Dar es salaam region and Kinondoni Municipal council. This work would not have been possible without the help of the in-charges and staff from the respective HIV care and treatment centers in Kinondoni Municipality and the research assistant. Our sincere gratitude goes to the study participants; their experience has given life to this research.

## Author Contributions

**Conceptualization:** Tasiana Njau, Bruno Sunguya, Sylvia Kaaya, Abebaw Fekadu.

**Data curation:** Tasiana Njau, Dorkasi L. Mwakawanga.

**Formal analysis:** Tasiana Njau, Dorkasi L. Mwakawanga, Agape Minja.

**Funding acquisition:** Tasiana Njau.

**Investigation:** Tasiana Njau.

**Methodology:** Tasiana Njau, Bruno Sunguya, Dorkasi L. Mwakawanga.

**Project administration:** Agape Minja.

**Supervision:** Bruno Sunguya, Sylvia Kaaya, Abebaw Fekadu.

**Validation:** Tasiana Njau, Abebaw Fekadu.

**Visualization:** Tasiana Njau, Bruno Sunguya, Abebaw Fekadu.

**Writing – original draft:** Tasiana Njau.

**Writing – review & editing:** Bruno Sunguya, Agape Minja, Sylvia Kaaya, Abebaw Fekadu.

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
