## [Decision Letter · Decision Letter 0]

7 Jun 2023

PONE-D-22-35584The unmet mental health needs of adolescents living with HIV in Eastern Tanzania: Experience of healthcare providers, adolescents, and caregiversPLOS ONE

Dear Ms. Tasiana

Thank you for submitting your manuscript to PLOS ONE. After careful consideration, we feel that it has merit but does not fully meet PLOS ONE’s publication criteria as it currently stands. Therefore, we invite you to submit a revised version of the manuscript that addresses the points raised during the review process.

We look forward to receiving your revised manuscript.

Kind regards,

Yared Abayneh Reta, MSC in ICCMH

Academic Editor

PLOS ONE

“TN Knowledges financial Support from the Centre for Innovative Drug Development and Therapeutics Trial for Africa (CDT-Africa),  https://www.cdt-africa.orga;  World Bank Africa Centre of excellence at Addis Ababa University, as funding for her Ph.D.  The opinions in this paper are those of the authors and do not reflect the views of CDT-Africa or The World Bank.”

5. Thank you for stating the following in the Acknowledgments/ Funding Section of your manuscript:

“TN Knowledges financial Support from the Centre for Innovative Drug Development and Therapeutics Trial for Africa (CDT-Africa), a World Bank Africa Centre of excellence at Addis Ababa University, as funding for her Ph.D.  The opinions in this paper are those of the authors and do not reflect the views of CDT-Africa or The World Bank.”

“TN Knowledges financial Support from the Centre for Innovative Drug Development and Therapeutics Trial for Africa (CDT-Africa),  https://www.cdt-africa.orga;  World Bank Africa Centre of excellence at Addis Ababa University, as funding for her Ph.D.  The opinions in this paper are those of the authors and do not reflect the views of CDT-Africa or The World Bank.”

6. Thank you for stating the following in your Competing Interests section: 

“The authors declare that they have no competing interests.”

Additional Editor Comments:

Dear Author, I would like to apologize for the delay in the review process. Finding reviewers has been quite difficult. Please read the review comments carefully and remember to respond to each remark and question point-by-point.

As part of your revised paper, kindly include the following:

- Include any set criteria's to include or exclude interviewees.

- Clearly put your objectives

- Specify which model of phenomenological qualitative study approach was employed

- Explain how you managed bias (did you use bracketing?)

- Since your research is qualitative, try replacing terms such as "high level of unmet..." with equivalent qualitative vocabulary.

Regards,

Reviewers' comments:

Reviewer's Responses to Questions

**Comments to the Author**

1. Is the manuscript technically sound, and do the data support the conclusions?

Reviewer #1: Yes

Reviewer #2: Yes

2. Has the statistical analysis been performed appropriately and rigorously? 

Reviewer #1: I Don't Know

Reviewer #2: I Don't Know

3. Have the authors made all data underlying the findings in their manuscript fully available?

Reviewer #1: Yes

Reviewer #2: No

4. Is the manuscript presented in an intelligible fashion and written in standard English?

Reviewer #1: No

Reviewer #2: Yes

5. Review Comments to the Author

Reviewer #1: The authors are to be congratulated for considering this important subject. Unmet mental health needs is the challenging problem not only in people living with HIV but also in the general population especially in low-income countries.

Please address the following issues regarding your manuscript.

1. The background of your manuscript needs rewriting. Because adolescents with HIV are not only risk for depression but they are also vulnerable for psychosis, mood spectrum disorder etc. Make it more inclusive.

2. You use purposive sampling approach to recruit the participants. Which type of purposive sampling you used? Specify it?

3. Since you have used human participants, instead of saying NA, you have to state the ethics statement from the online submission of your manuscript.

4. In your data availability statement, you have stated that all relevant data are with in the manuscript and its supporting information but your manuscript lacks supporting information. Please include the supporting information in your manuscript.

5. Please include a caption for table 1.

6. I note you have included a table to which you do not refer in the text of your manuscript. Please ensure that you refer to table 1 in your text.

7. Other problems are minor. The paper could be improved grammatically by using a native English speaker and lack of publication information for reference number 32.

Reviewer #2: 1. Why ethical review board of AAU needed? since the study was in Tanzania ? justify

2. It is interesting finding but would have sounded better if you had mixed with quantitative study design

3.Why only depression symptoms experienced ? why not psychotic symptoms ?or anxiety , how about substance use disorder

and related mental illness ??

4.samplze size did not look based on saturation , if so how did it happened to be equal among each participants ? by change ?justify

5.I believe that not having quantitative should be limitation of your study , you may do this as limitation

6.I recommend to to attach your questionnaires as annex ,others may refer it when needed.

6. PLOS authors have the option to publish the peer review history of their article (what does this mean?). If published, this will include your full peer review and any attached files.

Reviewer #1: **Yes: **Agmas Wassie Abate

Reviewer #2: **Yes: **Yacob Abraham Borie

---

## [Author Response · Author response to Decision Letter 0]

22 Aug 2023

We are attaching two versions of manuscripts as required; one with corrections highlighted and the second as a clean version. However, we apologize that the draft with track changes has not comprehensively included all changes made, as the first reviewer forgot to enable the track change functionality. We have, however, highlighted most changes that were made without track changes in yellow. We appreciate the time and expertise of the Editor and reviewers, that has to this point greatly improved the quality of our manuscript.

---

## [Decision Letter · Decision Letter 1]

2 Jul 2024

The unmet mental health needs of adolescents with HIV in Eastern Tanzania: Experience of healthcare providers, adolescents, and caregivers

PONE-D-22-35584R1

Dear Mrs. Tasiana

We’re pleased to inform you that your manuscript has been judged scientifically suitable for publication and will be formally accepted for publication once it meets all outstanding technical requirements.

Kind regards,

Yared Abayneh Reta, MSC in ICCMH

Academic Editor

PLOS ONE

Additional Editor Comments (optional):

Dear Authors,

As the Academic Editor, I apologize for the delay in the review of your manuscript. Finding suitable reviewers was quite challenging, and I appreciate your patience and understanding during this time. Please review the manuscript for any final comments before submission.

Thank you for your cooperation and patience.

Best regards,

Reviewers' comments:

Reviewer's Responses to Questions

**Comments to the Author**

1. If the authors have adequately addressed your comments raised in a previous round of review and you feel that this manuscript is now acceptable for publication, you may indicate that here to bypass the “Comments to the Author” section, enter your conflict of interest statement in the “Confidential to Editor” section, and submit your "Accept" recommendation.

Reviewer #2: All comments have been addressed

Reviewer #3: All comments have been addressed

Reviewer #4: All comments have been addressed

2. Is the manuscript technically sound, and do the data support the conclusions?

Reviewer #2: Yes

Reviewer #3: Yes

Reviewer #4: Yes

3. Has the statistical analysis been performed appropriately and rigorously? 

Reviewer #2: I Don't Know

Reviewer #3: Yes

Reviewer #4: Yes

4. Have the authors made all data underlying the findings in their manuscript fully available?

Reviewer #2: Yes

Reviewer #3: Yes

Reviewer #4: Yes

5. Is the manuscript presented in an intelligible fashion and written in standard English?

Reviewer #2: Yes

Reviewer #3: Yes

Reviewer #4: Yes

6. Review Comments to the Author

Reviewer #2: No comment , I am satisfied. This important for policy makers to focus on mental health issues.

It would be better if you could add quantitative parts to support your finding's figuratively. You may add or do it in future in accordance with current findings.

Reviewer #3: It would have been better if the writers change their title to “unmet mental health needs related to depression among AWHIV in eastern Tanzania:experience of adolescents, health care providers and caregivers”, to make it more specific and to avoid any confusion while reading the paper for the readers.

Reviewer #4: I would like to thank the authors for their nice work as well as for incorporating the feedback into the first version of the manuscript.

I would also like to appreciate the constructive comments of the previous reviewers.

For the authors:

It is not clear whether the study was conducted on all the adolescents attending the HIV care and treatment centers or focused on adolescents with both disorders only (HIV and mental illness). How adolescents with HIV and mental illness are identified and accessed is not clear (if that were the case).

Adolescents who experienced symptoms of mental illness and/or suicidal wishes were not referred for further intervention for their sufferings (not stated under the ethics section).

7. PLOS authors have the option to publish the peer review history of their article (what does this mean?). If published, this will include your full peer review and any attached files.

Reviewer #2: **Yes: **Yacob Abraham Borie

Reviewer #3: No

Reviewer #4: No

---

## [Editor Report · Acceptance letter]

11 Jul 2024

PONE-D-22-35584R1 

PLOS ONE

Dear Dr. Njau, 

I'm pleased to inform you that your manuscript has been deemed suitable for publication in PLOS ONE. Congratulations! Your manuscript is now being handed over to our production team.

Kind regards, 

on behalf of

Dr. Yared Abayneh Reta 

Academic Editor

PLOS ONE